# BLOCK-ATTENTION FOR EFFICIENT PREFILLING

**Dongyang Ma**[*]
Tencent
dongyangma@tencent.com

**Yan Wang**[*†]
Tencent
yanwang.branden@gmail.com

**Tian Lan**
lantiangmftby@gmail.com

## ABSTRACT

We introduce Block-attention, an attention mechanism designed to address the increased inference latency and cost in Retrieval-Augmented Generation (RAG) scenarios. Traditional approaches often encode the entire context in an auto-regressive manner. Instead, Block-attention divides retrieved documents into discrete blocks, with each block independently calculating key-value (KV) states except for the final block. In RAG scenarios, by defining each passage as a block, Block-attention enables us to reuse the KV states of passages that have been seen before, thereby significantly reducing the latency and the computation overhead during inference. The implementation of Block-attention involves block segmentation, position re-encoding, and fine-tuning the LLM to adapt to the Block-attention mechanism. Experiments on 11 diverse benchmarks, including RAG, ICL, and general domains, demonstrate that after block fine-tuning, the Block-attention model not only achieves performance comparable to that of full-attention models, but can also seamlessly switch between the block and full attention modes without any performance loss. Notably, Block-attention significantly reduces the time to first token (TTFT) and floating point operations (FLOPs) to a very low level. It only takes 45 ms to output the first token for an input sequence with a total length of 32K. Compared to the full-attention models, the TTFT and corresponding FLOPs are reduced by 98.7% and 99.8%, respectively. Additionally, in Appendix A, we elaborate on how Block-attention is applied in Game AI scenario and the substantial potential benefits it entails. We strongly suggest researchers in the gaming field not to overlook this section. [1]

## 1 INTRODUCTION

Retrieval-Augmented Generation (RAG) (Li et al., 2022; Lan et al., 2023) is a crucial technology for mitigating knowledge hallucination in Large Language Models (LLMs). By utilizing retrieval technology, LLMs can seamlessly access passages stored in external databases, grounding their responses in the content of these passages. To the best of our understanding, RAG has emerged as the most effective method for infusing specific domain knowledge into LLMs in real-world scenarios.

However, everything has two sides, and RAG is no exception. Generally, for each user query, in order to ensure that a passage with the "correct knowledge" is successfully recalled, it is a common practice to retrieve multiple passages—typically between 5 to 30 in most scenarios (Kwiatkowski et al., 2019; Joshi et al., 2017). These passages are then incorporated into the input prompt for the LLM. As a result, the inference efficiency decreases significantly due to the increased sequence length of this extended input prompt. Specifically, the inference latency, measured as the time to first token (TTFT), is considerably higher for a RAG-LLM compared to a non-RAG LLM (Li et al., 2023; Zhu et al., 2024).

---

[*] Equal Contribution
[†] Corresponding Author
[1] Codes, datasets and model weights have been publicly available at `https://github.com/TemporaryLoRA/Block-attention`.

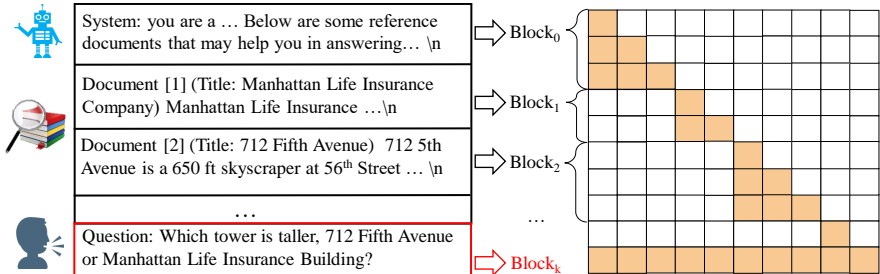

Figure 1: The Block-attention Masks

Given that the passages in the external databases might have been computed, it is natural to restore their KV cache for fast inference and avoid re-computing these passages. Nonetheless, for autoregressive LLMs, the KV states are inherently context-dependent, which means the KV states for the same passage will vary in different contexts. As a result, when encountering an unseen query, the model must undertake a complete re-encoding of the KV states to ensure an accurate prediction.

In this paper, we propose the Block-attention method, which reduces the TTFT and computation FLOPs of RAG-LLMs to a level nearly equivalent to that of non-RAG LLMs, while fully maintaining the same accuracy level. As shown in Figure 1, the main idea of Block-attention is to divide the entire input sequence into several blocks. Each block independently calculates its KV states through full-attention, without any attention to other blocks. Only the final block is able to attend other blocks, *i.e.,* the user query is able to attend all the retrieved documents in the previous blocks. When utilizing Block-attention in RAG scenarios, we may achieve substantial benefits by defining each passage as a single block and caching its KV states in the memory for further reuse.

The implementation of Block-attention can be easily achieved through the following steps: 1) Encode all blocks except the last one separately; 2) calculating the positional encoding for each based on its position in the input prompt; 3) Integrating these blocks to compute the KV states for the final block. However, the primary challenge in utilizing Block-attention is that the LLMs have not been exposed to such an attention mechanism during their training, leading to difficulties in accurately interpreting the input prompt[2]. In our preliminary experiments, we attempted to directly implement Block-attention in the LLMs without updating any parameters. Unfortunately, this approach resulted in a substantial decrease in performance, with the average accuracy of Llama-3.1-Tulu-3-8B-SFT on four RAG benchmarks falling from 66.1% to 49.9%. [3]

To address this challenge, we implemented a fine-tuning process for the LLMs to adapt to the Block-attention mechanism. Our experiments demonstrated that, after approximately 500-1000 fine-tuning steps, the Block-attention model achieved a full recovery of its original accuracy across all scenarios, impressively increasing from 49.9% to 70.0%. This outcome underscores the Block-attention LLMs' capability to uphold inference accuracy while significantly enhancing inference efficiency in RAG scenarios.

We conduct comprehensive evaluations of the Block-attention mechanism across 11 diverse benchmarks, including RAG, ICL(In-Context Learning), and general domains. Experimental results demonstrate that after fine-tuning, the average accuracy of the Block-attention model on the benchmarks can remain comparable to *Llama-3.1-Tulu-3-8B-SFT*. In terms of efficiency, we counted the TTFT and FLOPs to the first token (FLOPs-TFT) of the Block-attention model when the length of user input is 50 and the total length of the input sequence gradually increases. We found that the longer the total length, the more obvious the improvement of Block-attention on efficiency. When

---

[2]Although some studies have proposed training-free frameworks (Gim et al., 2024; Merth et al., 2024) utilizing parallel context encoding and KV caching to achieve passage-level KV cache reuse, they encounter significant degeneration issues in practical applications. Our experimental results also demonstrate that the results of these methods are far from satisfactory.

[3]We also proposed some training-free solutions to mitigate this huge gap. If someone is interested in switching to Block-attention in an on-the-fly manner, please refer to Zhang et al. (2024) for the detail.

the length of the input sequence reaches 32K, the TTFT and FLOPs-TFT of the Block-attention model are 1.3% and 0.2% of that of the full-attention model, respectively.[4]

## 2 BLOCK-ATTENTION

### 2.1 MAIN IDEA

Let $\mathcal{S} = \{s_0, s_1, ..., s_n\}$ represents the input sequence, where each $s$ represents a token. We denote the KV states associated with $\mathcal{S}$ as $\mathcal{K} = \{k_0, k_1, ..., k_n\}$ and $\mathcal{V} = \{v_0, v_1, ..., v_n\}$, respectively. For an auto-regressive model $\Theta_{LLM}$, since the computation of the KV states is dependent on their preceding tokens, when a text block $\{s_i, ..., s_j\}$ changes to a new text block $\{s'_i, ..., s'_m\}$, for the new sequence $\mathcal{S}' = \{s_0, ..., s'_i, ..., s'_m, s_{j+1}, ..., s_n\}$, its KV states become $\mathcal{K}' = \{k_0, ..., k'_i, ..., k'_m, k'_{j+1}, ..., k'_n\}$ and $\mathcal{V}' = \{v_0, ..., v'_i, ..., v'_m, v'_{j+1}, ..., v'_n\}$. It is evident that although only one block $\{s_i, ..., s_j\}$ has changed, due to the auto-regressive nature of LLMs, the KV states of all subsequent blocks must be re-encoded.

Our research focuses on exploring a novel attention mechanism named Block-attention. This mechanism is designed in such a way that it only requires the re - computation of the text blocks that have been altered between two input sequences. As a result, it can achieve an outcome that is equivalent to that obtained from fully re - encoding the entire sequence.

As illustrated in Figure 1, the essence of Block-attention is to divide the input sequence $\mathcal{S}$ into several independent blocks. Each block autonomously calculates its KV states through self-attention, without considering other blocks. The final block, however, has the unique capability to integrate information from preceding blocks. A primary advantage of this method is the modular independence it provides: when a block $b_i$ is updated to $b'_i$, re-encoding only the KV states of the affected block $k_{b'_i}, v_{b'_i}$, and those of the final block $k_{b_k}, v_{b_k}$, is sufficient to obtain the updated KV states.

To develop a Block-attention LLM capable of precise inference, we must tackle three challenges:

1) How do we segment blocks?

2) How should the positional encoding be calculated for each block?

3) How can the LLM be adapted to the Block-attention mechanism?

These issues will be addressed in detail in Sections 2.2, 2.3, and 2.4, respectively.

### 2.2 BLOCK SEGMENTATION

The primary principle of block division is to segment semantically independent parts of the prompt into separate blocks. In RAG scenarios, since the retrieved passages are originally mutually independent, it is natural to divide them into different blocks. Therefore, let's go back to the left part of Figure 1, where we allocate each passage to a single block and designate the user's query as the final block. This principle extends to other scenarios as well. For example, in the context of code generation tasks, a function may be treated as one block; in multi-turn dialogues, each turn could be segmented into an individual block; while in In-context Learning (ICL), each demonstration naturally forms a self-contained block (we will validate the efficacy of Block-attention in ICL scenarios in our experiments). In this paper, our primary focus is on the application of Block-attention in RAG, with the exploration of other scenarios reserved for future research.

### 2.3 POSITION RE-ENCODING

The second problem is to re-encoding the positional information. Although the same passage may appear in multiple input prompts, its position generally varies. Therefore, when we attempt to reuse the KV states of a block, we need to re-encode its positional information. The process of re-encoding is simple and straightforward: taking the rotary positional encoding (RoPE) as an example, assume we wish to change the positional encoding of a block $b = \{s_i, ..., s_j\}$ to $b' = \{s_{i_\Delta}, ..., s_{j_\Delta}\}$, then we only need three steps:

---

[4]Given that the KV cache is already a mature and low-cost technology (Qin et al., 2024; Lee et al., 2021), in this paper we do not take the cost of KV cache into account.

1) For the token $s_i$, its positional encoding vector $f(s_i, i)$ is calculated using the following formula:

$$f(x_i, i) = \begin{pmatrix} \cos i\theta_0 & -\sin i\theta & \cdots & 0 & 0 \\ \sin i\theta_0 & \cos i\theta & \cdots & 0 & 0 \\ 0 & 0 & \cdots & \cos i\theta_{\frac{d}{2}-1} & -\sin i\theta_{\frac{d}{2}-1} \\ 0 & 0 & \cdots & \sin i\theta_{\frac{d}{2}-1} & \cos i\theta_{\frac{d}{2}-1} \end{pmatrix} \begin{pmatrix} p_0 \\ p_1 \\ \cdots \\ p_{d-1} \end{pmatrix} \tag{1}$$

2) We rotating $x_i$ counterclockwise by $i\theta$ degrees, to re-set its positional encoding to zero:

$$f(x_i, 0) = \begin{pmatrix} \cos i\theta_0 & \sin i\theta & \cdots & 0 & 0 \\ -\sin i\theta_0 & \cos i\theta & \cdots & 0 & 0 \\ 0 & 0 & \cdots & \cos i\theta_{\frac{d}{2}-1} & \sin i\theta_{\frac{d}{2}-1} \\ 0 & 0 & \cdots & -\sin i\theta_{\frac{d}{2}-1} & \cos i\theta_{\frac{d}{2}-1} \end{pmatrix} f(x_i, i) \tag{2}$$

3) Then, by performing a clockwise rotation of $(i_\Delta)\theta$ degrees, we obtain the final positional encoding vector:

$$f(x_{i_\Delta}, i_\Delta) = f(x_i, i_\Delta)$$
$$\begin{pmatrix} \cos (i_\Delta)\theta_0 & -\sin (i_\Delta)\theta & \cdots & 0 & 0 \\ \sin (i_\Delta)\theta_0 & \cos (i_\Delta)\theta & \cdots & 0 & 0 \\ 0 & 0 & \cdots & \cos (i_\Delta)\theta_{\frac{d}{2}-1} & -\sin (i_\Delta)\theta_{\frac{d}{2}-1} \\ 0 & 0 & \cdots & \sin (i_\Delta)\theta_{\frac{d}{2}-1} & \cos (i_\Delta)\theta_{\frac{d}{2}-1} \end{pmatrix} f(x_i, 0) \tag{3}$$

For the remaining tokens within block $b$, namely $s_{i+1}, \ldots, s_j$, we can re-encode their positional information in a similar manner. Although the formulas presented above may seem complex, the principle is quite straightforward: **first set the positional encoding to zero, and then rotate it to the updated position**. One might wonder why we do not simply rotate by $(i_\Delta - i)\theta$ degrees directly? The reason is to mitigate the potential for errors in updating positional encodings within practical applications: in the KV cache, the positional encoding of the initial token of each block is standardized to zero, and with only the updated positional index $i_\Delta$, we can readily determine their new positional encoding vectors as per Equation 3.

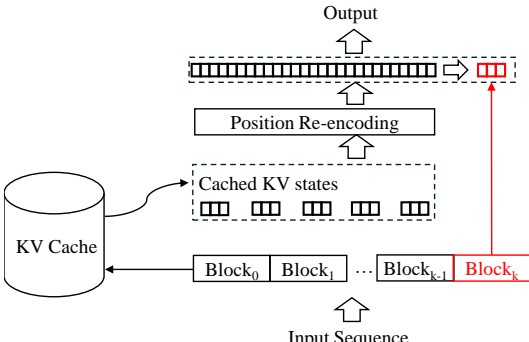

Figure 2: The Inference Pipeline of Block-attention Model

## 2.4 BLOCK FINE-TUNE

Due to the LLM's reliance on full-attention during the training phase, a direct switch to Block-attention during inference might result in a significant discrepancy between the training and inference states. Our preliminary findings indicate that introducing Block-attention without subsequent fine-tuning could precipitate a substantial decrease in performance, with the average accuracy dropping significantly from 67.9% to 48.0%. Adapting the LLM to Block-attention through fine-tuning, which we refer to as "**block fine-tune**," is quite straightforward. The only difference from the standard SFT process is the modification of the traditional lower-triangular attention mask matrix to the attention mask matrix depicted in the right part of Figure 1. With this masking matrix, tokens in all blocks except the last are restricted to attending only to information within their own block, thereby ensuring consistency between training and inference.

## 2.5 INFERENCE

In inference, the Block-attention model retrieves the KV states of blocks from the KV cache and concatenates them into the complete input KV states. The detailed process of the inference stage is depicted in Figure 2. Initially, we query and extract the KV states of the first $k-1$ blocks from

the cache. Then, based on the position of each block within the input sequence, we calculate their positional encoding using Equation 3. Finally, using the KV states of the first $k - 1$ blocks, we compute the KV states of the last block as well as the model output. In the RAG scenarios, the last block is the user query.

## 3 EXPERIMENTS

After the above analysis, there exist three concerns about the Block-attention method: 1) Can the Block-attention model achieve the same accuracy as full-attention in multi-block scenarios such as RAG and ICL? 2) Can the Block-attention model still adapt to the full-attention mechanism? 3) How much can the Block-attention mechanism improve the efficiency? The following experimental results will reveal the answers to these three questions. In Sections 3.5, we explored the answers to the first two questions and analyzed the accuracy of Block-attention models in 11 diverse tasks. Meanwhile, in Section 3.6, we demonstrate the efficiency of Block-attention in RAG scenarios.

### 3.1 DATASETS

**Train Dataset**  The first part of our training set comes from the SFT dataset of Tulu3[5]. For the samples in this dataset, we divide them into blocks according to three simple rules: 1) If it is a multi - turn sample, then we divide each turn (a user message and an assistant message) into an independent block; 2) The system message and the user message are assigned to two different blocks; 3) We directly use some newline characters, such as "\n\n", "—", "===", "\n\t", as block division labels. That is, when we encounter these characters, we divide the subsequent content into a new block. In this way, 23% of the Tulu3-SFT data can be used for block fine-tuning.

Another part of our training dataset is RAG samples. We randomly sample 20,000 instances from TriviaQA(TQA) (Joshi et al., 2017) and 2WikiMultiHopQA(2Wiki) (Ho et al., 2020) for fine-tuning models. Each training sample consists of (1) a question, (2) 10 passages retrieved from these two datasets using the Contriever toolkit[6], which identifies the 10 most relevant passages, and (3) an answer generated by Llama3.3-70B-Instruct based on the retrieved passages. The reason for using the Llama3 answer instead of the ground-truth answers is that the answer might not be present in our retrieved passages. This discrepancy could lead the model to overlook the content of the retrieved passages and generate outputs directly.

In the experiment, to maintain the full-attention ability of the model, we train the Block-attention model using both full-attention and Block-attention mechanism simultaneously.[7] In other words, all samples in the training set will be trained in both ways. For the full-attention baselines, only the full-attention method is used for training.

**Evaluation Dataset**  We evaluate the performance of our proposed Block-attention mechanism and baseline models on four widely-used RAG benchmarks: Natural Questions (NQ) (Kwiatkowski et al., 2019), TriviaQA (TQA) (Joshi et al., 2017), HotpotQA (HQA) (Yang et al., 2018), 2Wiki-MultiHopQA (2Wiki) (Ho et al., 2020), and NarrtiveQA (NQA) (Kočiský et al., 2017). Following Kandpal et al. (2023) and Liu et al. (2024), we use accuracy as our primary evaluation metric, judging whether any correct answers appear in the predicted output. To mitigate biases arising from output length, we set a maximum token limit of 200 for the output sequences.

In addition, we also evaluated the performance of the Block-attention model and the full-attention model in seven benchmarks of the general domain: MMLU (Hendrycks et al., 2021a), BigBench-Hard (BBH) (Suzgun et al., 2022), DROP (Dua et al., 2019), MATH (Hendrycks et al., 2021b), GSM8K (Cobbe et al., 2021), HumanEval (Chen et al., 2021), and IFEval (Zhou et al., 2023). [8] Please note that, among them, BBH, DROP, GSM8K, and MATH are ICL tasks with several independent in-context examples. We will divide each example into a separate block. For zero-shot

---

[5]https://huggingface.co/datasets/allenai/tulu-3-sft-mixture

[6]https://github.com/facebookresearch/contriever

[7]The details about our processed datasets can be found at `https://github.com/TemporaryLoRA/Block-Attention`.

[8]Since we do not have access to the OpenAI API, we are unable to conduct evaluations on datasets that require subjective assessment.

```
┌─ Input Prompt ─────────────────────────────────────────────────────────┐
│                                                                         │
│   You are an intelligent AI assistant. Please answer questions based on │
│   the user's instructions. Below are some reference documents that may  │
│   help you in answering the user's question. \n                         │
│ Doc-1                                                                   │
│   -Title: Manhattan Life Insurance Company \n                           │
│    Manhattan Life Insurance Company, incorporated on May 29, 1850, is   │
│    a life……\n                                                           │
│ Doc-2~n                                                                 │
│   - Title: New York Life Building \n                                    │
│     The New York Life Insurance Building, New York, located at 51       │
│   Madison Avenue, Manhattan, New York City, across……\n                  │
│                            …                                            │
│ User Query                                                              │
│   Please write a high-quality answer for the given question using only  │
│   the provided search documents (some of which might be irrelevant).    │
│   Question: Which tower is taller, 712 Fifth Avenue or Manhattan Life   │
│   Insurance Building?                                                    │
└─────────────────────────────────────────────────────────────────────────┘
┌─ Output ────────────────────────────────────────────────────────────────┐
│ The 712 Fifth Avenue tower                                              │
└─────────────────────────────────────────────────────────────────────────┘
```

Figure 3: The Inference Pipeline of Block-attention Model. The retrieved documents at the top have the highest relevance to the user query.

datasets like MMLU, HumanEval, and IFEval, the block attention model will directly switch to the full-attention mode.

## 3.2 INPUT FORMAT

The format of input prompt for all datasets follows Liu et al. (2024). For retrieved passages, we concatenate them in ascending order of retrieval score. An example is shown in Figure 3.

## 3.3 BASE MODEL & BASELINES

We implement the Block-attention mechanism on Llama-3.1-Tulu-3-8B-SFT (denoted as *Tulu3-SFT*)[9]. The reason why we choose this model is that it is a model that can be reproduced (They released their SFT data). By comparing different fine-tuning methods with the same data, we can easily demonstrate the effectiveness of Block-attention in different scenarios. After applying block fine-tuning, the model is denoted as *Tulu3-block-ft*. For comparison, we also implemented four baselines and three ablated models.

- **Tulu3-SFT:** Our base model, which also serves as the performance ceiling for Block-attention models in general tasks. We aim to make the Block-attention model's general performance approach this benchmark as closely as possible.

- **Tulu3-RAG:** Since *Tulu3-SFT* was not trained on RAG data, it demonstrates suboptimal performance in RAG tasks. To ensure fair comparison, we conducted supervised fine-tuning of *Tulu3-SFT* using our training data with full-attention mechanism. This model establishes the performance ceiling for Block-attention models in RAG scenarios. We similarly aim to have the Block-attention model's RAG performance approach this upper bound as closely as possible.

- **Tulu3-RAG-promptCache:** A baseline model that applies the PromptCache (Gim et al., 2024) to the *Tulu3-RAG* model. This enables training-free reuse of attention states across different prompts of large language models (LLMs).

- **Tulu3-RAG-Superposition:** Serving as a baseline model, it uses the Superposition method on the *Tulu3-RAG* model (Merth et al., 2024). This method allows the LLM to process input documents along parallel prompt paths. This parallel processing mechanism not only enhances the model's processing speed but also optimizes resource utilization by eliminating unnecessary computations.

- **Tulu3-block-ft-full**. An ablated model that switches the *Tulu3-block-ft* to the full-attention mode. We want to observe through the performance of this ablated model whether the Block-attention and the full-attention can coexist in the same model.

---

[9]https://huggingface.co/allenai/Llama-3.1-Tulu-3-8B-SFT

| Models | 2wiki | HQA | NQ | TQA |
|---|---|---|---|---|
| *Tulu3-SFT* | 62.0 | 68.4 | 58.6 | 75.7 |
| *Tulu3-RAG* | **73.2** | **74.8** | **61.5** | **75.8** |
| *Tulu3-RAG-Superposition* | 30.1 | 32.3 | 35.9 | 58.9 |
| *Tulu3-RAG-promptCache* | 32.4 | 31.6 | 44.4 | 61.8 |
| *Tulu3-block-ft* | **72.2** | 72.3 | **60.4** | 75.1 |
| *Tulu3-block-ft-full* | **73.6** | **75.2** | **62.2** | **76.2** |
| *Tulu3-block-ft-w/o-pos* | 68.9 | 69.9 | 59.2 | 74.4 |
| *Tulu3-block-w/o-ft* | 42.9 | 42.1 | 48.3 | 66.5 |

Table 1: Accuracy of different models on four RAG benchmarks.

| Task Type | General | | | ICL | | | |
|---|---|---|---|---|---|---|---|
| dataset | IFEval | HumanEval | MMLU | GMS8K | MATH | BBH | DROP |
| setup | 0-shot | 0-shot | 0-shot | 4-shot | 4-shot | 3-shot | 3-shot |
| *Tulu3-SFT* | 68.5 | 58.5 | **63.7** | 75.5 | **29.2** | **68.5** | 9.4 |
| *Tulu3-RAG* | 68.3 | **65.2** | 63.6 | 75.6 | 28.6 | **68.5** | 10.4 |
| *Tulu3-block-ft* | **70.0** | 59.1 | 63.0 | **75.7** | 28.8 | 65.3 | **14.4** |

Table 2: Accuracy of different models on seven general benchmarks. For the first three zero-shot benchmarks, the Block-attention will fall back to full-attention. For the subsequent four ICL datasets with few-shot examples, each sample will be divided into an independent block. Therefore, for a k-shot sample, it will be divided into k+1 blocks.

- **Tulu3-block-w/o-ft**. An ablated model that transitions the attention mechanism of *Tulu3-RAG* to Block-attention mechanism without any block fine-tuning. The outcomes of this model represent the lower bounds for the Block-attention model's effectiveness, given that it has not undergone any adaptation to Block-attention during the training phase.

- **Tulu3-block-w/o-pos**. Another ablated model that is also fine-tuned to adapt the Block-attention mechanism, while no additional position re-encoding operations described in Section 2.3 are conducted. This model will be used to evaluate the effectiveness of the proposed position re-encoding process.

## 3.4 EXPERIMENTAL SETUP

**Training and Inference**    All experiments are conducted using 8 NVIDIA H20 GPUs with following hyper-parameters: (1) learning rate $\alpha = 2 \times 10^{-5}$; (2) batch size $b = 64$; (3) epochs $n = 1$; and (4) 20 warmup steps. The DeepSpeed[10] and Flash-Attention (Dao et al., 2022) toolkits are utilized to accelerate our training procedure using bfloat16 format. Additionally, Flash-Attention is utilized for efficient inference of our fine-tuned models and baselines.

**Evaluation**    OpenCompass toolkit (Contributors, 2023) is used to evaluate the performance of models under general benchmarks and RAG benchmarks.

## 3.5 MAIN RESULTS

From the results in Table 2, we can draw four important conclusions:

1) It is not advisable to directly switch from full-attention to parallel prompt encoding methods including Block-attention, as it will lead to a sharp drop in accuracy. For instance, as can be seen from the experimental results of Tulu3-block-w/o-ft, removing the Block fine-tune process causes the Tulu3-RAG model to experience an average absolute performance decrease of 16.2% across all four RAG benchmarks. Additionally, the performance degradation of Tulu3-promptCache and Tulu3-Superposition is even more significant, which also indicates that the current parallel prompt encoding methods are still far from satisfactory. Through case studies, we found that these models have serious degeneration problems.

---

[10] https://github.com/microsoft/DeepSpeed

2) However, if we use the Block-attention mechanism in the fine-tuning stage, then the resultant model has comparable performance with its full-attention counterpart. In RAG scenarios, the performance gap between *Tulu3-block-ft* and *Tulu3-RAG* on four benchmarks is less than or equal to 1%. Meanwhile, *Tulu3-block-ft* significantly outperforms *Tulu3-SFT* on 2wiki and HQA. In the ICL scenario, *Tulu3-block-ft* also has an overall performance almost the same as or even slightly higher than that of the two full-attention models. This conclusion indicates that in RAG and ICL scenarios, it is completely feasible to replace full-attention with Block-attention, and there will be almost no performance loss.

3) Block and full attention can seamlessly transition. When we switch *Tulu3-block-ft* to the full-attention mode, we are pleasantly surprised to find that, whether in the RAG scenario (denoted as *Tulu3-block-ft-full*) or in the naturally single-block zero-shot scenario (IFEval, HumanEval, and MMLU), the Block-attention model can achieve performance comparable to or even slightly higher than that of the two strong full-attention baselines. This conclusion, together with the previous one, indicates that using both Block-attention and full-attention during training can lead to the acquisition of a model that is capable of seamlessly transitioning between the two attention mechanisms.

4) The position re-encoding operations are essential for the Block-attention Model. Removing it leads to a certain degree of performance drop—an average 2% decrease in accuracy on all RAG datasets. Additionally, in this case, the model occasionally experiences the problem of degeneration.

Finally, one may still be interested in knowing exactly how many training steps are needed for the model to adapt to the Block-attention mechanism. Therefore, we counted the average accuracy of *Tulu3-block-ft* and *Tulu3-block-ft-full* on four RAG benchmarks at different fine-tuning steps and plotted it in Figure 4. It can be observed that at the beginning stage of fine-tuning, there is a huge performance difference between the two attention modes. It makes sense because the model needs more training steps to adapt to the Block-attention manner. After about 800 training steps, the model can switch seamlessly between the two attention modes without any performance loss.

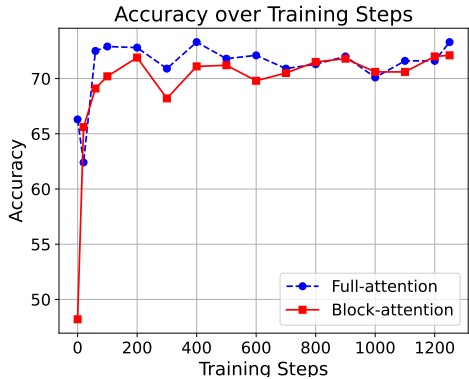

Figure 4: The accuracy of model checkpoint

### 3.6 INFERENCE EFFICIENCY OF BLOCK-ATTENTION

In the previous section, we already addressed our first concern: After fine-tuning, the Block-attention model can achieve similar or even better performance than the full-attention model. In this section, we focus on our third concern: How much can the Block-attention mechanism reduce the TTFT and FLOPs-TFT?

To quantify the effects of the Block-attention mechanism on the efficiency, we show in Table 3 the respective TTFTs and FLOPs-TFT of the *Llama3-block-ft* and *Llama3-vanilla-sft* when the KV states of all retrieved passages have been pre-computed and cached in memory. Obviously, the acceleration effect is gratifying: Once the length of the input sequence is 512 and the length of user input is 50, using Block-attention can reduce TTFT by 48% and reduce FLOPs-TFT by 90.1%. As the total length increases, the TTFT and FLOPs-TTF of the Block-attention model maintain an essentially unchanged trend. When the length reaches 32K, the acceleration effect reaches an astonishing 98.7%, and the consumption of FLOPs-TFT is even reduced by 99.8%. We may simply conclude the results as: *with greater text comes greater necessity for Block-attention.*

### 3.7 DISCUSSION

From the experimental results, we can figure out the effects of Block-attention on existing RAG applications: Under the existing technical paradigm, developers need to deliberately balance the trade-off between accuracy and latency. Therefore, they have to limit the number of retrieved passages to a certain number, such as 3 to 10. With Block-attention, the impact of the number of retrieved

| Prompt Length | 50 | 512 | 1K | 2K | 4K | 8K | 16K | 32K |
|---|---|---|---|---|---|---|---|---|
| TTFT-vanilla | 26 | 50 | 87 | 167 | 330 | 691 | 1515 | 3638 |
| TTFT-block | 26 | 26(48%) | 26(71%) | 26(84%) | 27(91%) | 29(95%) | 34(97%) | 45(98.7%) |
| FLOPs-TFT-vanilla | 7.5e+11 | 7.6e+12 | 1.5e+13 | 3.0e+13 | 6.1e+13 | 1.2e+14 | 2.45e+14 | 4.9e+14 |
| FLOPs-TFT-block | 7.5e+11 | 7.5e+11 | 7.5e+11 | 7.5e+11 | 7.5e+11 | 7.5e+11 | 7.5e+11 | 7.5e+11 |
| Reduction | - | 90.1% | 95.0% | 97.5% | 98.7% | 99.3% | 99.6% | 99.8% |

Table 3: The Time and FLOPs consumed by the first token of a user input with a length of 50 tokens under different total length of the retrieved passages

passages on inference speed will be greatly reduced, which will empower the developers to freely choose the number of retrieved passages that gives the optimal effect without any hesitation.

While our experiments in this paper focus on the RAG and ICL scenario using publicly available datasets, Block-attention's transformative potential extends far beyond these domains. Through validation across multiple internal tasks, we observed consistent efficiency gains. However, due to confidentiality constraints preventing data/code disclosure, we strategically selected the RAG scenario for reproducibility and benchmarking purposes.

Notably, Appendix A details how Block-attention addresses critical latency challenges in Game AI. We believe that Block-attention's true disruptive potential lies in **enabling real-time LLM agents** - a vision technically unattainable through full-attention due to the constraints of inference costs.

## 4 RELATED WORKS

### 4.1 CONCURRENT WORK

We were surprised to find that another ICLR 2025 submission, TurboRAG (Lu & Tang, 2025), independently proposed similar methods and reached similar conclusions as ours. They also proposed independent attention (which is the block attention in this paper) and reordered positions (the position re-encoding in this paper). Unfortunately, their work was not accepted by ICLR 2025. We sincerely hope that the community will also recognize them as one of the pioneers of Block-attention, and we hope that this similarity will not affect the publication of their paper in the future. Another study, DecoupledRAG (Dong et al., 2025), addresses the inefficiency issues in traditional Retrieval-Augmented Generation (RAG) methods by employing a cross-attention mechanism to directly inject external knowledge into the LLM's reasoning process.

Recently, two highly discussed papers in the sparse attention domain—NSA (Native Sparse Attention) from Deepseek (Yuan et al., 2025) and MoBA (Mixture of Block Attention) from Moonshot (Lu et al., 2025)—have independently proposed block-based attention mechanisms similar to our Block-attention. While these works are not strictly parallel to ours (their preprints were released five months after ours), we aim to clarify the distinctive contributions and orthogonality of our approach compared to these studies. After partitioning inputs into blocks, these methods employ a trainable block selection operation to filter out irrelevant blocks. The retained blocks are then concatenated and processed in an auto-regressive pre-filling manner. Our Block-attention, in the contrary, focuses on two novel aspects: 1) Parallel context encoding, and 2) Cross-prompt block KV cache reuse. Neither NSA nor MoBA supports these capabilities.

In our prior work Zhang et al. (2024), we demonstrated that even simple block selection mechanisms—when applied to parallel context encoding—can substantially improve model accuracy. This suggests the "Block-attention we proposed" addresses complementary optimization dimensions compared to the "Block-attention proposed by DeepSeek and Moonshot". In their future models, they may take our work as another pathway to significantly reduce their inference cost further.

### 4.2 RETRIEVAL-AUGMENTED GENERATION (RAG)

RAG is a widely used technique to improve generations of language models by using retrieved nearest-neighbor documents or passages as references, which typically involves two stages: retrieval

and generation. Before generation, retrieval finds most similar passages with the user query or the context, by using BM25 or dense retrieval model (Lee et al., 2021; Lan et al., 2023; 2024; Ma et al., 2025; Che et al., 2024). After collecting retrieved passages, there are numerous techniques to incorporate the knowledge during generation. Earlier works includes concatenation (Izacard & Grave, 2021), cross-attention (Borgeaud et al., 2022) and distribution interpolation Khandelwal et al. (2020). Some studies have also begun to attempt to directly use the retriever as text generator, that is, text generation is performed by selecting context-aware phrases from a collection of supporting documents (Lan et al., 2023; Cao et al., 2024).

Recently, LLMs becomes the most powerful paradigm for most NLP tasks, and simply concatenating all retrieved documents into the context of LLMs becomes the most simple and effective way for retrieval-augmented generation (RAG). For example, Self-RAG (Asai et al., 2023) leverage a critic model to decide which content in retrieved passages should be used during generation. As a specific application of RAG, tool learning is widely used to call external APIs to retrieve related passages from external database or tools to solve knowledge-intensive tasks (Schick et al., 2023).

### 4.3 Parallel Context Encoding

Some researches focus on individually and parallely process each documents, which is related to our work, such as SGLang (Zheng et al., 2024), FiD (Izacard & Grave, 2021), PCW (Ratner et al., 2023), PromptCache (Gim et al., 2024) and CacheBlend (Yao et al., 2024). Although SGLang could maintain the comparable generation quality as the original model, its reuse conditions are very strict, making it difficult to improve the inference efficiency. FiD is widely used for encode-decoder architecture, and not compatible for the decoder-only architecture of LLMs, since they need to concatenate the hidden states for decoder during inference. PCW focuses on extending the context window rather than efficient inference. The acceleration effect of PromptCache is similar to Block-Attention; however, as shown in our experiments, due to not properly handling positional encoding, its RAG performance is rather poor. CacheBlend introduces a trade-off between generation quality and KV cache reuse efficiency. Compared to SGLang (Zheng et al., 2024), CacheBlend achieves higher reuse efficiency and lower TTFT but at the cost of slightly reduced generation quality. On the other hand, compared to PromptCache, CacheBlend incurs overhead due to cross-attention recovery but delivers better generation quality.

From the above analysis, one may derive an intriguing conclusion: existing studies are forced to compromise either on generation quality or inference efficiency. Some works maintain high quality with low efficiency (SGLang), others prioritize high efficiency (PromptCache), while some attempt to balance both aspects (CacheBlend). In contrast, our proposed Block-Attention eliminates the need for such trade-offs through position re-encoding and block fine-tuning. It achieves both the same generation quality as the original model and the reuse efficiency comparable to PromptCache, without introducing additional overhead. More importantly, the resulting model even achieves seamless switching between Block-Attention and full-attention modes, significantly enhancing the flexibility of online services.

## 5 Conclusion

We introduce Block-attention to optimize the inference efficiency of LLM in RAG scenarios. Its essence lies in separately calculating the KV states of independent blocks in the input prompt. Block-attention enables us to pre-compute the KV states for all passages and cache them in memory, thus avoiding repeated computation of the KV states of the same passage during inference.

Our experimental results across various RAG benchmarks demonstrated the profound impact of Block-attention. We showed that Block-attention can maintain the original reasoning accuracy of LLM while significantly reducing its TTFT and FLOPs-TTF in RAG scenarios. The effectiveness of Block-attention becomes increasingly apparent as the number of passages increases or the frequency of retrievals increases. When considering the implementation of Block-attention in your applications, bear in mind this guiding principle: With greater text comes greater necessity for Block-attention.

## 6 ACKNOWLEDGEMENTS

We sincerely express our gratitude to our friends, JCY, ZZL, HWY, and QXT, for their insightful suggestions and unwavering support during the inception of this idea. Unfortunately, due to their affiliation with a confidential institution, we are unable to include them in the author list. We earnestly hope that they will eventually enjoy a more open research environment.

We also extend our heartfelt thanks to our colleagues at Tencent AI Lab—Xinting Huang, Tian Liang, Jiahao Xu, and Zhaopeng Tu—for their valuable advice and resource support during both the rebuttal stage and the camera-ready phase of this work.

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

## A   BLOCK-ATTENTION IN GAME AI

Before detailing the application of Block-attention in Game AI, we first outline the characteristics of LLM tasks in gaming scenarios. In such contexts, the LLM input typically consists of the game's current state, known as gamecore data in the industry. This data is structured as JSON with lengths ranging from thousands to hundreds of thousands of tokens. Different tasks produce varied outputs, but the input generally remains the gamecore data of the current frame. For example: for AI players like AlphaStar (Vinyals et al., 2019) and JueWu (Ye et al., 2020), their problem can be formulated as:

$$P(Player\ Action|gamecore).$$

For AI NPCs, their problem then becomes:

$$P(NPC\ Action|gamecore).$$

As for tasks like AI commentary, they can be defined as:

$$P(Game\ Comments|gamecore).$$

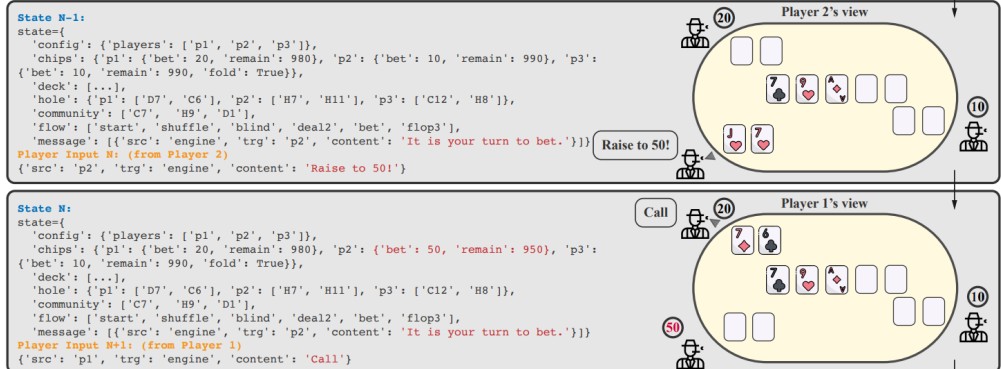

Figure 5: A case of Texas hold'em AI's gamecore data

As shown in the formulations above, most gaming tasks involve extremely long contexts (from several K to hundreds of K) but relatively short output sequences (tens to hundreds of tokens). If the AI must prefill data from scratch for every decision, the resulting excessive inference latency would render LLMs impractical for real-time applications. This explains why current MOBA and FPS games still rely on small-parameter models for online deployment.

To illustrate how Block-attention reduces TTFT in LLMs, consider a Texas Hold'em example. Figure 5 compares gamecore data at two consecutive states: State N-1 (previous) and State N (current). Notably, these states are nearly identical except for the value of *state['chips']['p2']*, which changes from {*'bet': 10, 'remain': 990*} to {*'bet': 50, 'remain': 950*}. This high inter-frame repetition (exceeding 99.5% based on our analysis) is common in games. Furthermore, the structured JSON format of gamecore data allows rule-based partitioning into independent blocks. By encoding only new or modified blocks during prefilling (instead of prefilling from scratch), we achieve equivalent accuracy to full prefilling while significantly reducing latency.

In an unreleased game (not yet available to the public), by dividing the game state into over 300 independent blocks, we achieved the same accuracy as full-attention mechanisms in these tasks, while cutting the average TTFT from 2,800ms to 100ms and reducing query latency from 3,000ms to under 300ms. Our Block-attention method aims to help AI game developers effortlessly create more games powered by advanced AI systems. We envision that Block-attention will empower AI researchers in the gaming industry to seamlessly develop more AI-centric gaming experiences.

