# OpenReview forum: "Block-Attention for Efficient Prefilling"
_ICLR.cc/2025/Conference — ICLR 2025 Poster_

### Official Review · Reviewer_MNR9 · 2024-11-03

**Soundness:** 3
**Presentation:** 3
**Contribution:** 3
**Rating:** 6
**Confidence:** 4

**Summary:**

This paper presents the Block-Attention mechanism aimed at optimizing inference efficiency in retrieval-augmented generation (RAG) scenarios. The authors assert that by modifying the attention calculation to process tokens in independent blocks, they can significantly enhance both accuracy and computational efficiency compared to traditional self-attention. The proposed method is evaluated against well-known RAG benchmarks, and the authors claim notable improvements in inference speed and resource consumption.

While the topic is pertinent, the contributions of the paper raise several concerns. The approach, although presented as innovative, relies on established principles of attention mechanisms without sufficiently distinguishing itself from previous work. Additionally, the experimental results, while promising, lack depth in exploring the nuances of the proposed method's effectiveness across varying conditions.

**Strengths:**

1. The paper is generally well-organized, with clear explanations of the Block-Attention mechanism and its proposed advantages.
2. The investigation into improving inference efficiency in LLMs through Block-Attention is timely.
3. The reported gains in inference speed and reduced computational load are compelling.

**Weaknesses:**

1. The concept of segmenting attention into blocks is not entirely novel and has been explored in various forms in the literature. The paper does not convincingly articulate how Block-Attention offers distinct advantages over similar approaches, such as sparse attention methods or hierarchical attention mechanisms.
2. The paper presents empirical results without adequately establishing the theoretical underpinnings of the Block-Attention mechanism. For example, the claim that "tokens in all blocks except the last are restricted to attending only to information within their own block" (Section 3.4) needs further elaboration on how this restriction impacts the model's representational capacity compared to full self-attention.
3.  The experimental section would benefit from a broader set of benchmarks and comparison against state-of-the-art methods beyond just self-attention.
4. The authors should discuss how the model performs under different conditions, such as varying levels of noise in the retrieved passages or the diversity of user queries.

**Questions:**

1. How do the authors justify the choice of block size in the Block-Attention mechanism, and what implications does this have for different types of queries?
2. Can the authors elaborate on how the performance of Block-Attention varies with input length and retrieval volume? Are there any thresholds beyond which the performance might degrade?
3. Given the reliance on fine-tuning, how do the authors plan to address potential overfitting issues, particularly in diverse RAG settings?

---

> ### Author Response · Authors · 2024-11-20
>
> Thanks for your valuable suggestions！
>
>  For **weakness 1: articulating  Block-Attention offers distinct advantages over similar approaches, such as sparse attention methods or hierarchical attention mechanisms**, we will explain in detail the differences between us and sparse attention methods or hierarchical attention mechanisms and update them in the manuscript:
>
> Differences between block attention and **sparse attention**: Our approach and sparse attention are two different and orthogonal routes. The goal of Sparse Attention is to achieve efficient RAG by compressing the amount of computation in attention. However, Block-attention does not conduct compression operations on the amount of attention computation (independently encoding different blocks will actually reduce the pre-filling cost to some extent, but this is just a side effect, not our main purpose). Instead, our main goal is to make the LLM reuse as many previous computation results as possible to achieve efficient RAG. The challenge of sparse attention is whether the compression operations can avoid losing important information, while the challenge faced by block-attention is how to reuse as many previous computation results as possible.
>
> As for the **hierarchical attention mechanism**, we have discussed the main differences with a representative work, "Parallel context windows for large language models" (lines 349-355): The goal of most hierarchical attention mechanisms is still to extend the context window to handle long-context that longer than the model's window size. When the input length does not exceed the context window, adopting this mechanism will significantly reduce the model's accuracy.
>
> **weakness 2: Lack of theoretical analysis**
>
> We sincerely apologize for the insufficient theoretical analysis. The block-attention mechanism stems from some intuitive observations within real-world applications. Hence, we placed greater emphasis on empirical analysis in the paper. In accordance with your suggestion, we will incorporate more extensive elaboration on the block-attention mechanism in the next version of the paper.
>
> **Weakness 3: comparison against state-of-the-art methods beyond just self-attention.**
>
> In fact, we have already made comparisons with two state-of-the-art methods, namely PCW and PromptCache respectively. The results of PCW are even lower than those of the weak baseline, and the results of PromptCache are exactly the same as our weak baseline. To avoid redundant content in the table, instead of putting them in the table, we described them in words from line 349 to 355.
>
> In addition, following the suggestion of Reviewer wUEZ, we have supplemented the comparison with two related works. Please refer to our **response to Reviewer wUEZ** for the detailed results.
>
> **Question 1: How do the authors justify the choice of block size in the Block-Attention mechanism, and what implications does this have for different types of queries?**
>
> Actually, we didn't configure a hyperparameter like "block size" for the model. The quantity of blocks is determined by the number of semantically independent parts in the prompt. For instance, in the RAG scenario introduced in this paper, each retrieved passage will be segmented into an individual block. If 100 passages are retrieved, then the prompt will be partitioned into 102 blocks (1 system block + 100 passages + 1 user query block).
>
> Moreover, we can also give a brief introduction to the principles of block division when handling other types of prompts (in non-RAG scenarios). In the coding scenario (such as Copilot), each class and function will be segmented into an independent block. In the few-shot learning scenario, the examples within the prompt will be placed into an independent block. There are also some more general rules. For instance, we can utilize symbols like "```" and "\n\n" as block delimiters. Thus, in the training set of our LLM, the blocks are already predefined by these pre-defined rules along with manual labeling.
>
> During the Inference phase, the block division becomes even more straightforward. Apart from the aforementioned rules, those long text blocks that frequently reappear in different prompts will be defined as an independent block. We can readily obtain these high-frequency blocks through log analysis.

---

> > ### Author Response · Authors · 2024-11-20
> >
> > **Question 2: Can the authors elaborate on how the performance of Block-Attention varies with input length and retrieval volume? Are there any thresholds beyond which the performance might degrade?**
> >
> > Sure no problem. First, we'd like to introduce our experience of using block attention in real-world applications (not just in this paper). The performance of block-attention will only decline significantly (be lower than the vanilla-sft model) under one circumstance: when the number of blocks in the Inference stage is greater than the maximum number of blocks in the Training stage, and the length of the last block is very short, the performance of block-attention will be inferior to that of self-attention. We can intuitively understand it as the fact that since the number of tokens with global attention (the last blocks) is too small, the models fail to fully "integrate" the knowledge from different blocks during the prefill stage. In practice, we have a strategy to address the potential drop caused by this issue: we may set a hyperparameter max_blocks = n, and then only the first n passages will use block-attention, and the subsequent documents will switch back to self-attention.
> >
> > In this paper, since most of the datasets have only 10 associated passages, we didn't conduct this experiment. After you raised this suggestion, we implemented an additional experiment on NaturalQuestion-Open: we changed the number of retrieved documents (only in the inference stage) as well as the number of blocks with global attention, and observed the performance changes of the block-attention model.
> >
> > |Num. of passages|3|5|10|15|20|30|
> > | :--- | :--- | :--- | :--- | :--- | :--- | :--- |
> > |Accuracy (Global block num. =1)|54.5	|55.9	|56.2	|56.6	|54.1	|52.4|
> > Accuracy (Global block num. =3)	|53.9|	55.8|	56.9|	57.2|	55.7	|54.7|
> > Accuracy (Global block num. =5)	|54.0	|55.7	|57.1|	57.3|	56.8	|57.2|
> >
> > We may easily observe that: When the maximum number of blocks in the training stage is 10, the model can extrapolate to at most 15 blocks in the inference stage, and any more than that will lead to a decline in accuracy. However, if we increase the number of blocks with global attention (for example, by switching the last 5 blocks back to self-attention), we can successfully "reverse" this downward trend.
> >
> > **Question 3: Given the reliance on fine-tuning, how do the authors plan to address potential overfitting issues, particularly in diverse RAG settings?**
> >
> > Please refer to our general response to all the reviewers, where we have provided a detailed clarification of some queries related to "additional fine-tuning" and "generalizability & avoid overfitting". Once we adopt the block-attention mechanism during the SFT stage of the foundation model, the overfitting issues will no longer exist.
> >
> > Thus, our plan is not confined to technology alone. We will also keep on encouraging certain open-source LLM teams, such as Llama, Qwen, DeepSeek, and Hunyuan, to incorporate the block-attention mechanism into their future models. Stay tuned!

---

> > > ### Comment · Reviewer_MNR9 · 2024-11-26
> > >
> > > Thank you for he response. All my concerns have been addressed, and I recognize the contributions of this paper. Therefore, I will raise my score.

---

> > > > ### Author Response · Authors · 2024-11-27
> > > >
> > > > Thanks for your response!

---

### Official Review · Reviewer_1DTJ · 2024-11-03

**Soundness:** 3
**Presentation:** 3
**Contribution:** 3
**Rating:** 6
**Confidence:** 4

**Summary:**

This paper tackles the efficiency challenges in RAG. The key observation is that different paragraphs retrieved in RAG are independent to each other. Therefore, it is possible to perform independent self-attention within each paragraph, and only allow the final question query to attend to all previous passages. This will dramatically improve TTFT in long-context RAG. The authors identify two important steps to recover accuracy after converting traditional self-attention to block attention (proposed in this paper): positional re-encoding and fine-tuning. Their final results show no loss of accuracy and significant speedup in long-context RAG applications.

**Strengths:**

1. The idea is simple and reasonable: different retrieved passages are not necessarily related to each other so the attention mask can be sparsified.
2. Positional re-encoding is an elegant way to solve inconsistencies in positional encodings.
3. Finetuning results show no loss of accuracy and significant speedup over self-attention baselines.

**Weaknesses:**

1. The method requires fine-tuning, which limits its scalability.
2. The evaluation seems to be a bit weak.
- For example, the method trains on TQA/2Wiki's training set and evaluates on their validation sets. The results on these two benchmarks are not zero-shot and are not very representative.
3. It's unclear what is the overhead associated with positional re-encoding.
4. Can you construct some examples where different retrieved passages are related to each other? In this case, will the proposed method fail? For example, you might retrieve several chapters in a textbook, where later chapters are dependent on earlier chapters.

**Questions:**

Please respond to my questions in "Weaknesses".

===

Post rebuttal: my concerns have been resolved and I will keep my score.

---

> ### Author Response · Authors · 2024-11-20
>
> Thanks for your valuable suggestions！
>
> For weakness 1: The method requires fine-tuning, which limits its scalability. We have elaborated on the issue of “additional fine-tuning” in the general response to all reviewers. We believe that for any new attention mechanism, this cost is inevitable. However, as long as the post-training of some open-sourced foundation model support the block-attention manner, then we no longer need this additional fine-tuning process.
>
> **Weakness 2: The evaluation seems to be a bit weak.**
>
> We quickly conducted an evaluation on NarrativeQA (https://huggingface.co/datasets/deepmind/narrativeqa) and this is the result:
>
> ||NarrtiveQA|
> | :---        | :---        |
> Llama3-vanilla-sft|59.8|
> Llama3-block-ft|60.5|
>
>
> We will supplement more comperhensive experiments in the next version of the paper.
>
> **Weakness 3: It's unclear what is the overhead associated with positional re-encoding.**
>
> The cost associated with positional re-encoding is so low that it can be disregarded (< 0.001 ms). As a result, we did not conduct an analysis on the corresponding overhead in the paper.
>
> **Weakness 4: Examples where different retrieved passages are related to each other? **
>
> First of all, based on our observations in real-world applications and experiments, the model that has undergone block-fine tuning can easily handle such cases.
>
> Then, following your suggestion, we selected a case from the novel Harry Potter and the dataset HPD [1] to demonstrate the effectiveness of the proposed method in this scenario. Specifically, we took a consecutive piece of text from Harry Potter Book 6-Chapter 30 as the knowledge background (about 3,000 tokens) and then randomly divided it into 10 blocks. In this case, different retrieved passages are related to each other, but the block-attention model will encode each passage independently. Based on this background, we raised two questions for both the Llama3-vanilla-sft model and the Llama3-block-ft model.
>
> Due to space limitations, we have updated the detailed case in Appendix A of the manuscript. By carefully observing the output results of the two models, it can be found that for Question 1: What will Harry Potter's next actions be after the funeral? Both the Llama3-vanilla-sft and the Llama3-block-ft provided good answers. And for the more difficult Question 2 which requires reasoning based on multiple passages: Which characters are currently hated by Harry Potter? Among them, who is the one he hates the most? Unfortunately, the Llama3-vanilla-sft failed to output the correct answer, while the Llama3-block-ft successfully identified the relationships between Harry Potter, Lord Voldemort, and Severus Snape and output the correct answer.
>
> The most crucial reason why the Llama3-block-ft model can handle such cases lies in the full-attention design of the last block. Intuitively, this full-attention block is capable of discerning the semantic relationships (if any) between blocks and properly leveraging them to generate the output. We once attempted to apply block-attention to the last block as well and found that the accuracy would plummet directly to the level of the no-RAG model.
>
> We are confident that these comments will address the majority of your concerns about our study. Should you have any follow-up questions, we will be more than glad to provide you with the answers.
>
>
> [1] Large Language Models Meet Harry Potter: A Bilingual Dataset for Aligning Dialogue Agents with Characters.

---

### Official Review · Reviewer_wUEZ · 2024-11-04

**Soundness:** 2
**Presentation:** 2
**Contribution:** 2
**Rating:** 5
**Confidence:** 4

**Summary:**

The paper proposes Block Attention for retrieval augmented generation (RAG). The idea is to confine attention within each retrieved document (a.k.a. block) and only let the query attend to all documents. In this way, the KVs for each document can be cached and reused if the document re-appears in multiple queries. The method also proposes to explicitly correct RoPE positional encoding before reusing the cached KVs. After fine-tuning, this technique can retain the accuracy of the baseline while speeding up the inference due to smaller attention span and avoiding the KV recomputation thanks to the block caching.

**Strengths:**

- The method is simple and easy to implement: It only involves limiting the attention computation within each document in RAG scenarios and caching the KVs for each block for document re-use as proposed in PromptCache [Gim et. al, 2024]. The newly introduced RoPE rotation correction is also straightforward to implement.
- Local attention and prompt caching are sound techniques and suitable for RAG applications.
- The paper is clear and easy to follow.

**Weaknesses:**

My primary concern is regarding the paper's contribution compared to the existing works and the need for a better experimental analysis to understand where this method stands among previous methods that share very similar ideas or how it improves them. The authors can find more detailed comments and suggestions below.

- Contribution and limitations compared to existing works

To me the main contributions of the paper to improve the efficiency of RAG can be summarized as 1) limiting the attention within each document 2) caching the KVs of the documents to reuse them 3) Adjusting the rotation in the RoPE position encoding based on the position of the reappearing cached document. Points 1 and 2 have been already proposed in prior works. E.g. PromptCache [Gim et. al. 2024] proposed the same techniques to speed up the RAG applications. This makes contribution 3, the main new addition to the existing works. However, according to the experiments reported in the submission, the RoPE position correction does not lead to any improvement over PromptCache (lines 353-356). To my understanding this makes fine-tuning the only source of improvement over PromptCache, which is an expected gain which comes with known challenges of such a fine-tuning.

I want to also point the authors to [1] below that is a closely relevant prior art which is currently missing from the draft. [1] also addresses the three aforementioned points above. Namely, it proposes limiting attention to attention blocks, caching the KVs for the documents to re-use them, and addresses the position encoding problems by overlaying the documents as parallel edges without requiring any fine-tuning. I would suggest the authors add a discussion on the advantages of the proposed method compared to [1] and also add controlled experimental comparisons to better highlight the contributions.

[1] Superposition Prompting: Improving and Accelerating Retrieval-Augmented Generation, Merth et. al.

* Experimental Results

The experimental section needs to be revised to clearly highlight the main sources of improvements and contributions between the proposed method and existing prior methods (under same settings, e.g. with or without fine-tuning). Based on the submission, comparisons with similar approaches are missing (see [1] above), additional speed up on top of PromptCache (if any) is not verified, and it is not clear to me if anything other than the fine-tuning is the source of accuracy improvement over PromptCache. The authors can find more detailed comments and questions regarding the experiments in the following section.

**Questions:**

- Additional questions and comments:

1) To my understanding, the main technical contribution of the method compared to the previous approaches is the proposal to explicitly re-adjust the positional encodings. What is the practical advantage of the proposed RoPE encoding adjustment? Does it lead to any improvement on the final RAG task quality metrics over PromptCache? From the current manuscript I did not find an ablation showing its importance to improve prior arts.

2) What is the comparison between the proposed method and PromptCache in terms of efficiency? How much does the proposed method improve PromptCache in terms of wall-time?

3) How does the method compare with respect to the Superposition Prompting paper above [1] both in terms of accuracy and efficiency?

4) The paper mentions fine-tuning on RAG-specific datasets necessary for the proposed approach. Expectedly, fine-tuning can improve the accuracy on the specific tasks at hand, but it is also important to maintain the generalizability of the pre-trained models. This can be difficult especially when the training recipes are not available, or they are instruction fine-tuned on non-available data. How does the performance of the reported models change on common benchmarks such as MMLU and HumanEval after the suggested fine-tuning?

5) The positional encoding re-adjustment is only discussed for RoPE. As a suggestion for further improvement, it would be nice to have a more thorough analysis on different positional encoding approaches.

---

> ### Author Response · Authors · 2024-11-20
>
> Thanks for your valuable suggestions and concerns. Most of your concern is highly insightful, indicating that you are an expert in this field. We will provide a detailed explanation and address your concerns in the subsequent comments.
>
> **Weakness: Contribution and limitations compared to existing works. & Question 1, 2, 3**
>
> We sincerely apologize for the lack of clear explanation regarding the differences from PromptCache in our paper, as well as the missing reference: Superposition Prompting: Improving and Accelerating Retrieval-Augmented Generation. Please allow us to utilize this rebuttal opportunity to further clarify the distinctions and comparisons between our work and these two relevant works.
>
> **Differences and Comparisons with PromptCache:** In a real RAG scenario, PromptCache can be implemented in two ways. The first is the standard implementation method described in its paper, that is, not handling positional IDs at all. The accuracy of the PromptCache re-implemented according to this method is lower than that of block-attention (Please refer to the second and third rows in the table below). And the results of PromptCache after fine-tuning are the same as those of block-ft-w/o-pos, which are also lower than those of block-ft models (Please refer to the fourth and fifth rows in the table below).
>
> | | tqa|	2wiki	|nq|	hqa|
> | :---        |   :---  |       :--- |:--- |:--- |
> Llama3-block-w/o-ft	|62.7|	48.1|	44.3|	39.6|
> Llama3-promptCache	|60.8|	37.1|42.4|32.9|
> Llama3-Superposition|	57.9|	35.4	|37.9|	33.4|
> Llama3-block-ft	|73.0|	73.3|	56.2|	68.5|
> Llama3-promptCache-ft	|70.8	|66.8|	52.9|61.4|
>
> Another implement manner for PromptCache is exactly the one implemented in our paper. In this regard, we will reuse cached passages only on the condition that the positional IDs remain undisturbed. For instance, if the position ID of the preceding token of Doc A is 1000, then the cached KV states will be utilized only when the start positional ID associated with the cached Doc A is greater than 1000. When PromptCache is implemented in this manner, we can guarantee that there will be no reversed positional IDs, and its performance is exactly the same as that of block-w/o-ft. However, as we explained in footnote 6, the efficiency of PromptCache implemented in this way is considerably lower than that of the block-attention method. Quantitatively speaking, if the same amount of memory as that used by block-attention is employed, PromptCache can only reduce the TTFT (Time to First Token) by 50%. If it is desired to achieve the same TTFT as that of block-attention, PromptCache needs to consume k times the amount of memory (where k is the number of retrieved documents).
>
> Furthermore, following the suggestion of Reviewer 1DTJ, we have uploaded two examples in the Appendix A (The authors undertake to refrain from cherry-picking). Through these two instances, you can gain an intuitive understanding of the distinctions  between  Llama3-block-ft, Llama3-vanilla-sft, Llama3-block-w/o-ft, and PromptCache. The models after applying PromptCache will encounter serious fluency issues due to the disorder of positional IDs. These issues won't be accurately reflected in a research paper's automatic evaluation, but obviously, they are unacceptable in real-world scenarios.
>
> In the next version of our work, in order to prevent such confusion, we will update the implementation approach of PromptCache to the first type and incorporate the results of the above-mentioned table into the paper.
>
> **Differences and Comparisons with Superposition Prompting: ** The main idea behind SuperPosition prompting is to allocate passages to parallel paths for processing, thus reducing the inference cost. Its potential shortcoming is that each path can only attend to a single passage, whereas some questions might necessitate integrating information from multiple passages to be answered.
>
> Upon applying SuperPosition prompting to Llama3-vanilla-sft, the accuracy turns out to be lower than that of Llama3-block-w/o-ft and Llama3-PromptCache. The detailed results can also be seen in the table above.
>
> Furthermore, given that the decrease in the effectiveness of this method stems from the fact that each path can only attend to a single passage, it is quite challenging to enhance its accuracy even through further fine-tuning
>
> **Question 4:  the generalizability issue**
>
> Please refer to our general response to all the reviewers, where we have provided a detailed clarification of some queries related to "additional fine-tuning" and "generalizability & avoid overfitting". The model trained in our experiments will surely experience a decrease in generalizability, for we only use 2wiki and TQA as the training data. However, please note that this decrease is caused by our experimental setup rather than the block-attention mechanism

---

> > ### Author Response · Authors · 2024-11-20
> >
> > *Question 5: positional re-encoding for other positional encoding methods*
> >
> > Considering that the majority of open-source LLMs are implemented based on RoPE, our paper solely presents the position re-encoding based on RoPE. In the subsequent version of the paper, we will integrate a more in-depth and comprehensive analysis of various positional encoding methods.
> >
> > Your main concerns primarily arise from the distinctions and comparisons between block attention and related works, including PromptCache and Superposition Prompting. Since we have furnished detailed explanations and comparisons in this comment, we sincerely hope that you will consider adjusting your score. Should you have any follow - up questions, we will be more than glad to provide you with the answers.

---

> ### Comment · Reviewer_wUEZ · 2024-11-26
> **Follow up comments**
>
> I want to thank the authors for answering my questions.
> - Regarding [1], I am a bit confused by your explanation. Comparing Figure 1 in your submission, and Figure 3b in [1], I see no difference in attention patterns and based on that [1] should be also able to attend to the documents in the exact way similar to the proposed Block-Attention approach. After reading [1], I noticed that there the authors proposed an optional hyper-parameter "k" that can control how many documents the method attends to and to my understanding setting it to all the documents (that I suppose is the setting you are considering) should lead to the same config as the proposed method.
> - Regarding the comparison with PromptCache, thanks for the clarification. As you acknowledged, PromptCache can also achieve the same accuracy as the proposed approach. In this case, the main metric to look into should be efficiency and wall-time. However, I still couldn't find a head to head comparison in terms of wall-time between these approaches. Also as a follow up question, how much more memory PromptCache would require to achieve to the same accuracy and the same efficiency in practice on the datasets you considered?

---

> > ### Author Response · Authors · 2024-11-27
> >
> > Thank you for your comments. I'm delighted to engage in this follow-up discussion with you.
> >
> > **Regarding [1]**, please note that Figure 3.2 in this paper is merely a conceptual diagram, not the actual attention masks. Please directly refer to Section 3.2.1 of the original paper. The author emphasizes:
> >
> > “We emphasize that this resulting attention pattern is a construct for visualization only—in reality, all calls to the LLM use fully dense attention, although on relatively smaller context lengths. Concretely, each of the dashed boxes in Figure 2 is a separate LLM call.”
> >
> > Consequently, as we previously stated, the main idea of [1] is to split **one multi-passages query** into **multiple single-passage queries**, which differs significantly from our approach. As for their hyper-parameter k, it is used to control the number of minimum retaining paths in each step of the iteratively pruning process. Even if they set k to all documents, what they do is still to decompose the original query into k single-document query in each step. The additional experimental results we've obtained also confirm this point.
> >
> > **Regarding PromptCache**: Sure, no problem. We'll provide the detailed information regarding memory usage. Within the experimental setting described in the manuscript (where each query consists of 10 passages), in order to achieve the TTFT and FLOPs-TFT mentioned in the paper on the Natural_Question_open dataset, the block-attention model is required to cache 36,100 passages. Each passage has an average length of 148 tokens, with an average memory usage of 18.47 MB, resulting in a total memory consumption of **650GB**.
> >
> > As for the PromptCache model, if it aims to reach the same level of accuracy and latency as the block-attention model, the total memory consumption will be 650GB × 10 = **6.5TB**.
> >
> > In addition, please note that in real-world applications, with the help of an infra-team and the Prefill-Decoding Disaggregating framework, the actual memory consumption required is 1 to 2 orders of magnitude lower than the aforementioned two values.
> >
> > I hope my reply can address your concerns, and I'm also looking forward to further discussions with you.

---

> ### Comment · Reviewer_wUEZ · 2024-11-28
> **Post-rebuttal response**
>
> Regarding the discussion on comparison with PromptCache, I acknowledge that the main improvement of the proposed modification to the PromptCache algorithm is on the memory side and in terms of accuracy and latency, PromptCache can perform the same with or without the proposed positional ID adjustments. I suggest the authors to clarify this in the revised version of the paper and change the messaging to convey this summary.
>
> Re comparison with [1], their *Question* gets attended to each *Block*, and then the *Response* gets to attend to all the *Blocks* and the *Question* itself, as a result they still implement Figure 3.2 (although through multiple LLM calls) and at the end their response is generated based on multiple *Blocks* if they are all relevant to the question and the response. So your statement that "some questions might necessitate integrating information from multiple passages to be answered"and they can't do that is to me incorrect. The only difference that I can see here is that in your approach the *Question* attends to all the *Block*s at the same time, and for them the *Question* attends to each *Block* separately and the aggregation is postponed to the *Response* generation phase. However, this lets them to significantly parallelize the computation and reduce the latency compared to the proposed Block-Attention. A careful study is required to understand the benefits and shortcomings of each of these approaches both in terms accuracy and latency.
>
> I want to thank the authors for answering my questions. In summary, my main concern regarding contributions and limitations compared to the existing works mostly remains. However, authors' responses clarified the main advantage over the PromptCache baseline and I increased my score accordingly. I encourage the authors to modify the paper to highlight and focus on their main advantage over PromptCache (memory saving) and add discussions to delineate the differences, and contributions in terms of accuracy, memory and latency to the existing works.

---

> > ### Author Response · Authors · 2024-11-29
> >
> > Thank you for accepting our clarification. We will carefully analyze the differences and connections between our work and related works in the next version of the paper.

---

### Official Review · Reviewer_2J6u · 2024-11-05

**Soundness:** 4
**Presentation:** 3
**Contribution:** 3
**Rating:** 6
**Confidence:** 3

**Summary:**

This paper introduces the Block-Attention method, an efficient approach tailored for RAG scenarios that enhances inference efficiency while preserving performance through fine-tuning. Block-Attention operates by dividing the input sequence into multiple independent blocks, each calculating its key-value (KV) states separately via self-attention. Only the final block can attend to previous blocks, allowing the user query to reference all prior retrieved documents. Evaluations on four RAG benchmarks reveal that Block-Attention, after fine-tuning, maintains comparable accuracy to traditional self-attention models (e.g., 68.4% vs. 67.9% on Llama3) and even slightly outperforms in some cases (e.g., 62.3% vs. 59.6% on Mistral). The efficiency benefits, measured in TTFT and FLOPs, grow with input length; at 32K input length, Block-Attention’s TTFT and FLOPs reach just 1.3% and 0.2%, respectively, of those in standard self-attention models.

**Strengths:**

- Novel and practical idea
- easy to follow
- Simplicity: The solution is relatively simple to implement and can be integrated into existing LLM architectures.
- Impressive performance: this method reduces Time to First Token (TTFT) by up to 98.7% and FLOPs by up to 99.8%.

**Weaknesses:**

### Needs for finetuning:
The method requires additional fine-tuning, which might be resource-intensive for larger models. It would be better to explore a training-free approach with the proposed method.

**Questions:**

Only tested on relatively small models (7B-8B parameters) - would it work as well on larger models?

---

> ### Author Response · Authors · 2024-11-20
>
> Thanks for your valuable suggestions！
>
> We have elaborated on the issue of “additional fine-tuning” in the general response to all reviewers. We believe that for any new attention mechanism, this cost is inevitable. However, as long as the post-training of some open-sourced foundation model support the block-attention manner, then we no longer need this additional fine-tuning process.
>
> We apologize for forgetting to verify this aspect in our experiments. In real-world applications, it has been observed that the Block-Attention mechanism tends to show a decrease in effectiveness only when applied to very small-size models (those with fewer than 1 billion parameters). Nevertheless, on models with parameter counts varying from **1.5 billion to 130 billion**, it sustains an accuracy that is either comparable to or even better than that of the vanilla-sft models.

---

> > ### Comment · Reviewer_2J6u · 2024-12-03
> > **Official Comment by Reviewer 2J6u**
> >
> > Thank you for your comments and I decide to keep my initial rating.

---

### Author Response · Authors · 2024-11-20
**A general comment to all reviewer's concerns about "Needs additional fine-tuning" and "generalizability & avoid overfitting**

Thank you to all the reviewers for your valuable suggestions. I'm very glad to have the opportunity to engage in such an insightful discussion with you all on OpenReview. In this general response addressed to everyone, I'd like to briefly introduce the background of Block-Attention and the key points focused on in this research paper. I believe it can effectively address your concerns regarding regarding the issues of  "Needs additional fine-tuning" and "generalizability & avoid overfitting".

Firstly, any new attention mechanism requires fine-tuning, and Block-Attention is no exception. By way of analogy, transitioning from Alibi to RoPe also requires retraining the entire model from scratch. In real-world applications, the best approach to using Block-Attention is integrating it in the post-training stage of the foundation model. Specifically, for the LLM during this stage, we simultaneously adopt the self-attention mechanism and the block-attention mechanism to represent the training data. Regarding the division of blocks, some are manually divided during the annotation process, and others are divided according to rules. For example, in the RAG domain, each passage should be divided into one block; in the code domain, each function and class should be divided into one block; for few-shot tasks, each example can be divided into an independent block. The resultant model can adapt to both attention mechanisms simultaneously, and its results on common benchmarks are almost identical to those of the model without Block-Attention, without any performance loss. More importantly, after obtaining such a foundation model, we no longer need additional fine-tuning since the model has already completely adapted to the Block-Attention manner.

However, since there are currently no open-sourced foundation models that support block-attention, if we want to use the block-attention mechanism in downstream tasks to improve inference efficiency, we must conduct an additional fine-tuning. The adaptation cost in this aspect is inevitable. The difference only lies in whether this cost is borne during the training process of the foundation model or during the fine-tuning stage of the downstream tasks. Fortunately, this cost is not high. We don't need to change the attention-mechanism in the pre-training stage. It can be achieved simply by adjusting the attention masks in the downstream fine-tuning.

In this paper, due to confidentiality reasons, we are unable to release our internal SFT and Preference Optimization data. Therefore, in this submission, for the sake of re-implementation considerations, we focus on the RAG (Retrieval-Augmented Generation) scenario and only use publicly available dataset to train the model to verify the effectiveness of block-attention. The model trained in this way will surely have a decline in generalizability. However, please kindly note that this decline is caused by our experimental settings rather than by block-attention. We will also keep on encouraging certain open-source LLM teams, such as Llama, Qwen, DeepSeek, and Hunyuan, to incorporate the block-attention mechanism into their future models. Stay tuned!

---

### Meta-Review · Area_Chair_Dybj · 2024-12-22

**Metareview:**

a) The paper proposes Block Attention for retrieval augmented generation (RAG) with multiple documents (blocks). In this work, each document is assumed to be independent from the others, therefore, it is possible to reduce computation by performing independent self-attention within each block, and only allow the final question query to attend to all previous blocks. In this way, the KVs for each document can be cached and reused. The method introduces an explicitly correction of the positional encoding and fine-tuning, to retain the performance of the baseline model, but much faster inference.

b) The paper is well presented and easy to follow. The method is simple and easy to implement. Positional re-encoding is an elegant way to solve inconsistencies in positional encodings. The reported gains in inference speed and reduced computational load are compelling.

c) The concept of separating attention into blocks for RAG is not novel. It is already considered in works such as PromptCache and Superposition Prompting. In particular PromptCache seems very similar to the proposed idea, but missing the adjustment of the positional encoding and the fine-tuning.

d) The final decision for this paper was not easy. The paper has many similarities with PromptCache. Therefore the proposed idea is not novel and this should be clearer in the main paper. The main contributions are re-positional encoding and fine-tuning, which bring the method to the performance of the full self-attention, but with reduced computation. However, fine-tuning requires data and has the risk of over-specializing the model, losing generalization. Authors show results for in-domain and out-domain performance and the model seems to perform comparable to full self-attention also on out-domain. In addition, authors point out that block-attention could be used on the post-training phase, which would avoid the need of fine-tuning, although they did not provide results on that probably due to the training cost and time, but I do not see any reason why it should not work.
Considering all of that, I believe that the proposed ideas of re-positional encoding for improving PromptCache by itself can be quite important for RAG and deserves publication as it can produce similar retrieval performance as full-attention but with a fraction of the computational cost. Said that, the authors should do a good job on adapting the paper to reviewers comments and this discussion and make clearer what are the actual contributions compared with previous work.

**Additional Comments On Reviewer Discussion:**

Rev. 2J6u considered the main drawback the need for fine-tuning of the model. After authors rebuttal, rev. maintained their score of 6.

Rev. wUEZ pointed out the strong similarities of the paper to previous work. After a detailed discussion with authors, the comparative with related work is clearer. The main contribution of this work is the re-positional encoding and fine-tuning, that, when applied jointly can bring an important improvement on the retrieval and with performance comparable to full self-attention with a fraction of time. The final score of rev. is 5.

Rev. 1DTJ had an initial score of 5 because of some doubts and unclear parts in the paper. Authors did a good job in answering and rev. increased their score to 6.

Rev. MNR9 had a similar evaluation of the paper, with doubts that were clarified with the rebuttal. Thus, their final score was increased to 6.

After reading the review, my final decision was not clear. So, as suggested by the SAC, I also reviewed the paper. The paper is well written and the proposed idea of re-positional encoding for the blocks with fine-tuning is quite important for improved retrieval results. The main drawback is the fine-tuning, but it could be avoided if the method is integrated in the post-training of a model and therefore I would assign a score of 6.

Overall, I agree with rev. wUEZ analysis, but as expressed by rev. MNR9 in the internal discussion, I still consider that the proposed contribution is enough for acceptance as I believe that it could be impactful for improved RAG systems. However, authors should improve their presentation of the contributions with a correct analysis of related work in the final version of the paper.

---

### Decision · Program_Chairs · 2025-01-22

Accept (Poster)